# Closed-Loop Microreactor on PCB for Ultra-Fast DNA Amplification: Design and Thermal Validation

**DOI:** 10.3390/mi14010172

**Published:** 2023-01-10

**Authors:** Panagiotis Skaltsounis, George Kokkoris, Theodoros G. Papaioannou, Angeliki Tserepi

**Affiliations:** 1Institute of Nanoscience and Nanotechnology, National Center of Scientific Research (NCSR) “Demokritos”, Patr. Gregoriou Ε’ and 27 Neapoleos Str., 15341 Aghia Paraskevi, Greece; 2School of Medicine, National and Kapodistrian University of Athens (NKUA), 75 Mikras Asias Str., 11527 Athens, Greece

**Keywords:** polymerase chain reaction (PCR), closed loop, microreactor, lab-on-chip (LoC), printed circuit board (PCB), point of care (PoC), computational analysis

## Abstract

Polymerase chain reaction (PCR) is the most common method used for nucleic acid (DNA) amplification. The development of PCR-performing microfluidic reactors (μPCRs) has been of major importance, due to their crucial role in pathogen detection applications in medical diagnostics. Closed loop (CL) is an advantageous type of μPCR, which uses a circular microchannel, thus allowing the DNA sample to pass consecutively through the different temperature zones, in order to accomplish a PCR cycle. CL μPCR offers the main advantages of the traditional continuous-flow μPCR, eliminating at the same time most of the disadvantages associated with the long serpentine microchannel. In this work, the performance of three different CL μPCRs designed for fabrication on a printed circuit board (PCB) was evaluated by a computational study in terms of the residence time in each thermal zone. A 3D heat transfer model was used to calculate the temperature distribution in the microreactor, and the residence times were extracted by this distribution. The results of the computational study suggest that for the best-performing microreactor design, a PCR of 30 cycles can be achieved in less than 3 min. Subsequently, a PCB chip was fabricated based on the design that performed best in the computational study. PCB constitutes a great substrate as it allows for integrated microheaters inside the chip, permitting at the same time low-cost, reliable, reproducible, and mass-amenable fabrication. The fabricated chip, which, at the time of this writing, is the first CL μPCR chip fabricated on a PCB, was tested by measuring the temperatures on its surface with a thermal camera. These results were then compared with the ones of the computational study, in order to evaluate the reliability of the latter. The comparison of the calculated temperatures with the measured values verifies the accuracy of the developed model of the microreactor. As a result of that, a total power consumption of 1.521 W was experimentally measured, only ~7.3% larger than the one calculated (1.417 W). Full validation of the realized CL μPCR chip will be demonstrated in future work.

## 1. Introduction

In recent years, there has been a rapid expansion of biomedical applications based on microfluidics technology that allows the development of microscopic devices for handling liquid samples. The basic vision in microfluidics is the transfer of all analysis steps (sample preparation, mixing, reaction, separation, and detection), traditionally requiring a laboratory, to a miniaturized device, namely a lab-on-a-chip [1,2]. Lately, lab-on-chips have become extremely popular in the field of clinical diagnosis, mainly due to the increased need that has arisen for microdevices for personalized use, with the ability to provide fast and reliable diagnostic testing at the point where medical care is provided to the patient (point-of-care testing (POC)) [3]. POC microdevices promise to replace traditional diagnostic tests performed in microbiology laboratories, diagnostic centers, and hospitals, due to a number of advantages that they present. More specifically, such microdevices provide a significant reduction in both the cost of analysis and the time required to extract a result after sample collection, thus speeding up diagnosis and early treatment, as well as the possibility of carrying out the analysis by non-specialized personnel outside of central laboratories [4,5]. The very recent emergence of COVID-19 as a global pandemic has rendered the need for POC microdevices more than evident, in an effort to promptly diagnose and control the spread of the novel coronavirus [6,7,8]. In many cases of POC diagnostic microdevices (e.g., for pathogen detection), a necessary step for the analysis is the amplification of the genetic material for sensitive and reliable detection. For this reason, in recent years, much emphasis has been placed on the development of microreactors that accommodate the amplification of the genetic material.

Polymerase chain reaction (PCR) is the most common method used for DNA amplification and can produce a very large number of copies of a particular DNA fragment, starting from a small amount of the genetic material. PCR works with repeating thermal cycles, which typically consist of three steps (denaturation, primer annealing, and extension). Each of these steps requires a different temperature (e.g., 95 °C, 55 °C, and 72 °C, see Figure 1A), with the overall procedure usually requiring between 20 and 40 cycles.

As is the case with most lab-on-chips, the transfer of PCR to the microscale promises potential advantages and new possibilities. These include reagent cost reduction, faster reaction times, portability and autonomy, high automation, and high integration in one microdevice. For these reasons, a significant effort is devoted to creating reliable and sensitive PCR-based microreactors (μPCRs). These advantages, however, often come with certain challenges, which often have to do with the difficulty of mass-producing elaborate systems on a microscale. Another common obstacle encountered is the difficulty of realizing easy-to-use accompanying devices for controlling the flow and monitoring amplified DNA accurately in such microdevices. In recent decades, many different μPCRs have been developed, in an effort to overcome the above challenges and keep the advantages of μPCRs. These μPCRs use a wide variety of materials and manufacturing methods, microchannel geometries, temperature protocols, heating mechanisms, sample movement mechanisms, and DNA detection and quantification techniques [9,10,11,12,13].

The basic types of μPCRs include time-domain and space-domain microreactors [6,9,10]. In the former, the sample is usually placed inside one or more microchambers, where the temperature is then cycled several times, according to the selected PCR protocol (Figure 1A). The process is repeated for the necessary number of cycles until the desired DNA amplification is achieved. So far, many different approaches have been followed for time-domain μPCRs, each one with its advantages and disadvantages. In general, it can be said that time-domain μPCRs tend to be more time and energy consuming (besides exceptions [14]), compared to their space-domain counterparts. In some cases, time-domain μPCRs can be also more expensive and difficult to fabricate [9,10,15,16,17]. In space-domain μPCRs, the sample flows into microchannels, passing through areas of constant temperatures (Figure 1B). In this way, the temperature change occurs as the sample moves in space. The main advantage of these microreactors is that only the liquid sample is subject to temperature change, in contrast to time-domain μPCRs, in which the entire chamber is heated and cooled. This significantly reduces the thermal inertia of the system, allowing faster temperature changes, reduced reaction times, and lower energy costs. Several different approaches of space-domain μPCRs have been developed in the last two decades, one of them being oscillatory μPCRs. In this type of μPCR, the sample is forced to move back and forth between two or more temperature zones [18,19,20]. However, the most typical case is the continuous-flow (CF) μPCR. CF μPCRs comprise typically three heaters (which determine the three different PCR temperature zones) and a serpentine channel that lies over the heaters. The sample is pumped into the serpentine channel, with the PCR cycles taking place as the sample is heated and cooled, passing over the different temperature zones. The serpentine channel consists of many repeating identical units, each corresponding to one PCR cycle [21,22,23,24,25,26,27,28,29,30,31,32].

Although CF μPCRs come with many advantages, some drawbacks have emerged as well with these types of microreactors. First, in order to achieve the desired flow of the DNA sample, CF μPCRs are generally based on the use of pumps, which tend to raise the system cost and complexity [9,10,16,28,33]. At the same time, due to the long channel length, a large pressure difference is developed that requires very strong sealing of the microchannel in order to avoid liquid leakage [34,35]. Further limitations of typical CF μPCRs include the fixed number of PCR cycles, dictated by the channel layout [21,36], as well as the relatively large reactor footprint [21,37].

To overcome most of the drawbacks of time-domain and CF μPCRs, an advantageous type of CF μPCR has been proposed, i.e., the closed-loop (CL) μPCR. In this type of microreactor, the DNA sample flows through a circular microchannel (closed loop), passing consecutively through the different temperature zones, in order to accomplish a PCR cycle (Figure 1C). When the sample has moved through the whole circular microchannel for the first time, one PCR cycle has been completed. Then, the sample repeats the same path through the circular microchannel for the second cycle and so on, until the desired number of cycles is achieved.

CL μPCR offers the main advantages of CF μPCR, namely rapid temperature changes, short reaction times, and reduced energy costs. However, due to the circular geometry of the microchannel, it offers additional advantages such as fabrication simplicity to fabricate and smaller footprint, compared to serpentine CF μPCRs. Moreover, it offers flexibility in the number of PCR cycles, because it allows the fluid sample to recirculate as many times as necessary, in contrast to the fixed number of cycles in a serpentine microchannel. Another problem, often encountered in CF μPCRs with serpentine microchannels, is the difficulty in maintaining a uniform temperature throughout the temperature zone area. Due to heat losses, the temperature in the middle of the reactor is often higher than at the edges. This can result in PCR cycles being performed at slightly different temperatures, depending on whether the PCR thermal cycle takes place at the beginning, middle, or end of the microchannel. This phenomenon becomes more pronounced for devices with a larger footprint. CL μPCR does not face such a problem, as the sample passes through the same circular microchannel for every PCR cycle, and therefore, there is no question about non-identical temperature cycles. Another benefit of the circular arrangement of the three temperature zones in CL μPCRs is that it allows the temperature of the fluid sample to drop as fast as possible from the denaturation to the annealing zone, without having to pass unnecessarily through the extension zone, as is the case in CF μPCRs with serpentine microchannels (see Figure 1B). This improvement diminishes the possibility of rehybridization of the denatured single-stranded DNA between the denaturation and annealing zones, thus increasing the efficiency of the PCR cycle.

Although CL μPCRs with a circular microchannel offer many important advantages over traditional serpentine CF μPCRs, they have not attracted yet strong attention from the scientific community. Few studies have explored seemingly similar ideas, such as spiral-microchannel-based μPCRs [38,39,40,41] or spinning disc platforms [42,43,44], but a closer look reveals that these microreactors operate in a fundamentally different way from CL μPCRs, as they do not implement a closed-loop continuous flow. Some studies have achieved true closed-loop continuous flow by means of free convective flow, where natural thermal convection is generated by changes in the sample density, caused by the different temperatures inside the microreactor [17,45,46,47,48]. This approach often leads to microreactors that are based on simple designs and operate without the use of external pumps. Nevertheless, free-convective-flow μPCRs tend to use relatively big sample volumes [49], while the PCR cycle speed is limited due to the use of buoyancy forces for sample circulation.

The idea of a CL μPCR with a circular microchannel has been successfully applied in the past by Sun et al. [15,50,51]. In these studies, the microchannels are formed in a methyl polymethacrylate (PMMA) substrate. The circulation of the sample is achieved with the use of a ferromagnetic oil, which is inserted into the microchannel along with the reaction sample and cannot be mixed with it. The ferromagnetic oil is then forced to move by an external permanent magnet, which in turn moves with the aid of a micromotor. The ferromagnetic oil is, therefore, used as a plug, pushing the rest of the liquid sample and forcing it to flow through the circular microchannel. The heating is performed with Peltier-type thermal elements, not integrable in the PMMA chip. CL μPCRs have also been implemented in an older study [52], where sample circulation is achieved by using magnetohydrodynamic actuation.

It is evident that CL μPCRs with a circular microchannel offer many benefits, compared to other types of μPCRs, and Sun et al. [15,50,51] already showed that this type of microdevice can successfully work. However, the full potential of CL μPCRs is far from reached yet, and in fact, CL μPCRs can be improved by changes in materials and an optimized microreactor design. The aim of the current work is the upgrade of closed-loop microreactors through a) the use of a printed circuit board (PCB) as a substrate material and b) a detailed and improved design for a very fast and effective PCR. First, PCBs allow microheaters to be integrated into the microreactor, thus minimizing the total size of the microreactor, as well as the distance between the microheaters and the microchannel, therefore ensuring faster heat transfer. Second, a novel two-layer microheater design is proposed to optimize the PCR protocol by reducing the non-functional time in every PCR cycle. At the same time, the use of a PCB makes the microreactor fabrication process compatible with the established PCB industry, thus allowing for low-cost, reliable, reproducible, and mass-amenable fabrication, therefore enhancing the commercialization prospects of such a microreactor. Third, the best-performing microreactor is sought through a computational study, the first detailed one for this type of microreactor, to the best of our knowledge. The temperature distribution and the residence times at each thermal zone are calculated by a 3D heat transfer model, and the effects of microheaters’ design and operating conditions on the speed of the PCR are investigated. The low time limits in which a CL μPCR with a circular microchannel can operate are sought. Then, the best-performing microreactor is fabricated on a PCB substrate. Finally, the results obtained from the computational study are compared with the actual temperature distributions measured, by means of a thermal camera, on the surface of the fabricated microreactor.

## 2. Materials and Methods

### 2.1. PCR Protocol

Because polymerase can operate at a specific speed [53,54], a minimum amount of time is required for the complementary DNA strands to be formed during the extension step, making this step the rate-limiting step in the PCR cycle. As it is evident, this amount of time is directly related to the specific polymerase kit that is used in the amplification reaction, as well as to the length of the DNA fragment to be amplified. For the needs of this study, we considered a typical length of DNA fragment with 120 base pairs (bp) and a polymerase extension speed of 67 bp/s, given by product manufacturers [53,54]. From the above values, it follows that the minimum time required for the extension step should not fall below 1.8 s. Taking into account that when applying PCR at the microscale, the desired time ratio of the PCR steps should be close to 1:1:2 for denaturation: annealing: extension [22,23], we are led to the conclusion that the minimum time periods required for each step are 0.9 s for denaturation, 0.9 s for annealing, and 1.8 s for extension. This means that, for this type of μPCR, each PCR cycle needs to be at least 4 s. Any additional time comes from the transition of the sample temperature between the PCR thermal zones. The temperatures that are opted for for the three PCR steps are 95 °C for denaturation, 55 °C for annealing, and 72 °C for extension. More information about the PCR temperatures can be found in Section 3.1.

### 2.2. Microreactor Geometry

In order to proceed with the microreactor design, some basic features need to be determined, apart from the selection of a circular microchannel and the use of PCB as a substrate that have been already mentioned. The first microreactor feature to be decided is the volume of the microchannel (and thus the liquid sample volume). In general, the volume of the microchannel should be minimized, in order to ensure rapid temperature changes. Nevertheless, an extreme reduction in the sample volume could compromise the commonly used, subsequent process of gel electrophoresis, for the off-chip evaluation of DNA amplification. For these reasons, a volume close to 10 μL is selected, which is nearly half of the volume used in other CL μPCR studies [50,51]. The second feature to be determined is coupled to the substrate material of the microreactor and refers to the type of the microheating elements. Serpentine-shaped resistive copper microheaters are selected to maximize the resistance value for faster heating performance, thanks to their easy manufacturability and integrability into PCB substrates. The third microreactor feature refers to the dimensions of the microchannel. The depth of the microchannel should compromise the requirements for small “fluid thickness” to secure temperature uniformity and reliable and reproducible fabrication on a PCB substrate. A depth of 100 μm is selected, easily realized with the use of photolithography technique or computer numerical control (CNC) machining. Consequently, a width of 1 mm coupled with a 15 mm radius for the circular microchannel (and thus a length of 94.25 mm) is selected, in order to obtain the desired volume of ~10 μL (exact volume 9.42 μL). These values allow for reliable and reproducible fabrication of the microreactor on a PCB substrate, while at the same time keeping its footprint small.

Figure 2 shows the design of the microreactor. It comprises a circular microchannel and a cover on a PCB chip with integrated microheaters. The chip has a circular disk shape, with an external diameter of 40 mm and a 20 mm diameter hole in the center. Three distinct temperature zones are shown, each zone corresponding to one of the three PCR steps. The circular microchannel is formed on the top of the PCB substrate and is concentric with the chip disk, allowing the sample to move through the different temperature zones. The microchannel dimensions are, as mentioned above, 100 μm, 1 mm, and 94.25 mm, for depth, width, and length, respectively. The chip is covered with a very thin transparent layer of polyolefin, which would potentially allow, in the future, for real-time (fluorescence) monitoring of the reaction products. The polyolefin layer is 50 μm thick, PCR compatible, and has a pressure-sensitive adhesive on its bonding side. It can be bonded to the PCB by hand or with the use of a laminator (at approximately 90 °C) [55]. The temperature zones are defined with the copper microheaters, integrated into the PCB. The microheaters are used only for the denaturation and extension temperature zones, which need active heating. The annealing zone is designed to let the sample cool naturally by releasing heat into the environment, and so there is no need for a microheater. The microheaters have dimensions of 25 μm and 100 μm for thickness and width, respectively. On top of each temperature zone and exactly under the circular microchannel, a solid copper layer is used to improve the uniformity of the temperature zone [55,56]. Through holes of elliptical shape were designed between the temperature zones, to ensure better thermal insulation and reduce thermal cross talk.

Three different microreactor geometries are compared in the computational study. The goal was to find the best one in terms of PCR functional time ratio (i.e., total residence time of DNA in desirable temperatures divided by total cycle time) and speed. The 3D geometries of the microreactors that are used in the computational study were designed in Autodesk Inventor [57]. For the design of the PCB chip that was finally manufactured, an open-source software program, namely KiCad (v5.1.0) [58], was used. The three microreactor geometries, all designed in PCB, follow the same basic design and bear the same circular microchannel, as shown in Figure 2. The design parameter is the area in which the 2 microheaters extend, this parameter defining, along with the volumetric flow rate and the heat generation rate at the microheaters, the residence time in each of the desirable temperature ranges of the PCR.

In Figure 3A–C, the three different microreactor geometries can be seen. The difference between them lies in the design of the microheaters, as well as in the area that each zone occupies. More specifically, in the first microreactor geometry, each temperature zone occupies a third (120 degrees) of the total circle, with the integrated microheaters being formed in one internal copper layer (Figure 3A). This geometry follows a relatively simple approach by dividing the whole circular channel into three equal parts, as was the case in previous studies [15,50,51]. In the second microreactor geometry (Figure 3B), an attempt was made to improve the performance of the microreactor by changing the area ratios of the temperature zones. Given that each PCR step requires different temperatures that need to be maintained for different time periods, there is no reason why all three temperature zones should occupy the same area. We decided to reduce the denaturation zone area from one-third to a quarter of the total circle, in order to gain extra space for the two other zones. The reasoning behind this decision was that the microreactors in this study were designed to change temperatures with the use of active heating and passive cooling. In other words, the sample is actively heated by passing over the heating areas, but when it cools down, it simply releases its heat into the environment. This allows us to increase the heat absorbed by the sample per unit length in the heating zones and especially the denaturation zone, saving space for the annealing zone and the slower cooling of the sample. At the same time, the extension temperature zone needs to be enlarged as well, as the extension step is the most time-consuming step of the PCR cycle, in protocols similar to 1:1:2.

The third microreactor geometry (Figure 3C) keeps the same ratio of the temperature zones implemented in the second geometry but has a different microheater design. More specifically, at the beginning of each temperature zone, the microheaters are formed in two copper layers, so as to provide more heat (power) to the specific areas, in order to achieve the desired temperature in the sample faster and diminish the transition time between two PCR steps. On the contrary, in the middle of each temperature zone, the thermal elements are formed in just one layer, thus delivering less heat, so that eventually the temperature of the sample inside each zone is kept as stable as possible and close to the desired value. This novel two-layer microheater design aims to optimize the PCR protocol by reducing the non-functional time in every cycle and is realized for the first time in a PCR microreactor. This third microreactor geometry is also depicted in 3D in Figure 2.

### 2.3. Mathematical Model

The mathematical model consists of the energy conservation equation at the steady state:(1)$$\rho C_{p}u\cdot \nabla T+\nabla \cdot \left(-k\nabla T\right)=Q$$
where *ρ* is the density, *C*_*p*_ is the heat capacity at constant pressure, **u** is the fluid velocity vector, *T* is the temperature, *k* is the thermal conductivity, and *Q* is the volume heat source. The first term on the left-hand side applies only in the fluid domain, while the term *Q* applies only in the microheaters, and it is coming from Joule heating. Uniform heat generation rates are considered in the microheaters.

All the volume inside the microchannel is assumed to be filled with the liquid sample (aqueous solution), which is considered as water for the purpose of the computational study. The flow of the liquid sample inside the circular microchannel is assumed to be plug flow for the computational study, meaning that the velocity of the fluid is constant across any cross-section of the microchannel and normal to the cross-section and that there is no boundary layer adjacent to the inner walls. This assumption is made consistently with a realistic approach for the sample movement inside the circular microchannel. Indeed, in most of the reported studies of CL μPCRs with circular microchannels so far, the sample circulation is achieved by means of a ferrofluid plug [15,50,51], which induces plug flow.

Convective and radiative heat losses are considered; the heat transfer coefficient is considered equal to 5 W/(m^2^K). The emissivity values are shown in Table 1, together with the thermophysical properties of the materials in the stack. The ambient temperature is set equal to 296.15 K.

The software used for the computational study is COMSOL Multiphysics, in which the geometries designed in the previous step (in Autodesk Inventor) are imported. To obtain a mesh independent solution, approximately three million elements were required for each of the three different microreactor designs. More information about the mesh independency can be found in Appendix A.

### 2.4. Microreactor Evaluation with a Computational Study

To evaluate the performance of the microreactors, we examined the residence time of the sample at each step of the PCR cycle, as was the case in the recent work by Kaprou et al. [26]. The term “residence time” refers to the period of time during which a particle of the fluid sample lies within the desired temperature range to perform one of the three steps of PCR with satisfactory efficiency. These desired temperature ranges for each PCR step will be referred to from now on as “functional temperatures”. Temperatures between 89 °C and 97 °C were considered functional for the denaturation step, where denaturation of DNA double strands is achieved with efficiency greater than 80% [55]. Higher temperatures than 97 °C should provide good efficiency as well but were not included in the PCR protocol in order to minimize the risk of evaporation or deactivation of the amplification cocktail. For the primer-annealing step, the desired temperature range could vary and is strongly dependent on the specific primers that are used for PCR. In this study, we assumed an optimal primer annealing temperature between 54 and 56 °C [59], and a temperature range between 50 °C and 60 °C was considered functional. Extension is traditionally performed at 72 °C [60], but in some commercial applications, temperatures from 68 °C [61] to 75 °C [62] have been reported. In this study, temperatures between 67 °C and 75 °C were considered functional for the extension step, a temperature range in which polymerase activity is maximized.

Residence times are calculated by initially calculating the volume of the liquid sample that lies in the functional temperature range (for a study at steady state) and then multiplying it by the PCR cycle time and dividing it by the volume of the microchannel. The calculation of residence times is better illustrated in Section 3.1 in the results section, where we can see the residence times of an imaginary particle flowing on a streamline in the center of the microchannel.

From the residence times, we can also calculate the functional time ratio by dividing the total residence time of the three PCR steps (sum of denaturation, annealing, and extension residence times) by the total PCR cycle time.

### 2.5. Fabrication of the Optimum Microreactor

The design that gave the best results in the computational study, in terms of optimal residence times, minimum time loss, and fastest PCR cycle, was fabricated in order to measure the actual temperatures on its surface and compare them with the results of the computational study. The chip fabrication was realized on commercially available PCB substrates at a PCB manufacturing company (*Eurocircuits LTD*, *Mechelen, Belgium*), according to our specifications and design. The thickness of the PCB was 1.68 mm, and the resistance values of the two microheaters were measured at R_D_ = 23.3 Ω and R_E_ = 28.2 Ω in room temperature (23 °C) for the microheaters of the denaturation and the extension zone, respectively. Τhe manufacturing cost did not exceed EUR 17 per chip unit (gross price including 21% VAT), but it could be reduced by manufacturing a larger quantity of chips (e.g., the cost is reduced to EUR 4 per chip unit for an order size of 100 chips). A custom-made temperature controller was implemented to adjust each heater’s temperature and maintain it at the desired set-point [26]. The evaluation of the temperature on the surface of the PCB chip was performed with the use of a thermal camera. A FLIR A300 model was used, with the addition of an external lens, for magnification of the image. This method was preferred over a thermocouple, as it allows the temperature profile of the whole chip surface to be monitored. To ensure that a stable temperature is achieved at the chip over time, which is essential for a reproducible PCR, the thermal camera was kept monitoring the chip for 30 min.

## 3. Results and Discussion

### 3.1. Temperature Distribution and Residence Times in the Microreactor

The temperature field at steady state and the residence times for each PCR step and for each microreactor design (see Figure 3) are calculated. These residence times depend on three factors: (a) the footprint of the microheaters, which changes in each microreactor design, (b) the thermal power produced by the microheaters, and (c) the volumetric flow rate, the latter being directly related with the cycle time of the PCR protocol. For each of the three designs, different cycle times (and thus volumetric flow rates) are examined; the aim is to seek the lowest time limits in which each microreactor can operate providing an extension residence time (ERT) of at least 1.8 s, which is the minimum acceptable ERT in this study (see Section 2.1 above). For all of the examined cycle times, the heat generation rates by the two microheaters are adjusted, by means of an iterative procedure, so that maximum residence times are achieved for the denaturation and annealing steps. The values of these rates can be seen in Appendix A.

Figure 4 shows the ERTs that occurred for the different tested cycle times for each microreactor. The tests stop when a microreactor accomplishes the desired ERT of at least 1.8 s, as only the fastest cycle times are of interest. It can be seen that for all three microreactors, the ERT grows proportionally to the cycle time, a fact that makes sense since a longer cycle time means that there is more time available for each separate step to take place. Most importantly, the fastest possible cycle times at which each microreactor can operate (i.e., keeping an ERT above the minimum 1.8 s) can be deduced from Figure 4. These cycle times are 7.4, 7.0 s, and 5.7 s for microreactors 1, 2, and 3, respectively.

It can be seen that the third microreactor is the fastest of the three and can complete one PCR cycle in 5.7 s, which is equivalent to a total reaction time (30 cycles) of less than three minutes (171 s). Nevertheless, an ERT greater than 1.8 s is not by itself a sufficient condition for the microreactor to work properly; residence times of the other two steps need to be above the minimum time limits as well.

In Figure 5, the temperatures across the three microreactors can be seen. The results presented in Figure 5 are computed for volumetric flow rates that correspond to the fastest cycle time of each microreactor as described above. This means that in Figure 5A–C, we can see the first, second, and third microreactors performing a PCR cycle in 7.4 s, 7.0 s, and 5.7 s, respectively. All of the presented temperatures are taken at a plane in the middle of the microchannel, just 50 μm below the polyolefin cover. In all microreactors, the thermal crosstalk effect can be observed between the temperature zones (thermal crosstalk refers to the phenomenon in which heat generated in a temperature zone affects the temperature of neighboring zones), which occurs due to the high-speed fluid movement that transfers heat from one temperature zone to the next. Although thermal crosstalk should be minimized in such microreactors, it does not, however, affect much the reaction, as it just causes a displacement of each PCR step toward the next temperature zone. This means that the actual position where a PCR step takes place, e.g., denaturation, is not directly above the microheater that corresponds to the temperature zone but a little displaced toward the next zone (in this case, toward the annealing zone).

The detailed temperature profile of the microreactor as a function of time is shown in Figure 6A, with the third geometry being used as a representative case. The functional temperature range for each PCR step that is mentioned in the Materials and Methods section is highlighted with red, blue, and green colors for denaturation, annealing, and extension, respectively. All the temperatures presented in Figure 6A are calculated on a path along the center of the cross-section of the microchannel. From this figure, one can also read the residence time periods, as the time during which the sample remains at the functional temperature ranges in each PCR step. It should be noticed that the time when the sample cools between the denaturation and annealing steps is not counted as residence time for the extension step, although the sample passes from the above-mentioned extension functional temperatures. The reason is that, in order for the extension step to work properly, it needs to take place right after the annealing step, when the primers have already been bound to the DNA single strands. Instead, during the cooling stage, the sample reaches the desired extension temperature but this time with the denaturation step having preceded. This means that, at this stage of the cycle, no primers are attached to the DNA single strands, and thus, the extension of DNA cannot occur. Not counting the cooling time as part of the ERT applies to all the residence times that are calculated for the extension step in this study.

The residence times of each PCR step separately and in sum, as well as the total time needed for the completion of a whole PCR cycle, are presented in Figure 6B, for each of the three microreactor geometries. Again, these results are computed for the same volumetric flow rates as in Figure 5 and refer to the fastest cycle times that could be achieved for each microreactor geometry, thus constituting a good criterion for the evaluation of the performance of each microreactor. As expected, the first microreactor geometry has (unnecessarily) the longest denaturation residence time, even longer than the extension residence time.

The second microreactor geometry has a shorter denaturation residence time, providing in exchange a small improvement in the total cycle time, compared to the first microreactor. These microreactor geometries give residence time ratios of 1.30:1:1.26 and 1.12:1:1.48 (first and second microreactor geometry, respectively), both of them being far from the desired 1:1:2 protocol. The most important disadvantage that appears in these microreactor geometries is that they exhibit slow transitions between the temperature zones, which leads to delays that come at the cost of a lower functional time ratio and longer cycle time. This problem is addressed in the third microreactor geometry with the addition of a second copper layer for the microheaters. The second microheater layer is essentially located at the beginning of the two heating temperature zones (denaturation and extension) as can be seen in Figure 3C and Figure 7A (the sample flows counterclockwise). This way, the sample is heated more rapidly when entering each of the heating temperature zones, when intense heating is needed in order to minimize the transition times. In the middle of the temperature zones, the second microheater layer ceases to exist, so the temperature is kept constant.

The best results are obtained from the third microreactor geometry, both in terms of the desired protocol (1:1.06:1.54, which is closer to the desired 1:1:2 ratio) and the minimized total cycle time. At this point, it should be clarified that the residence times of each PCR step are not proportional to the areas of the corresponding thermal zones. It is true that, at first glance, it seems absurd to aim for a 1:1:2 protocol while having different thermal zone ratios. However, as was mentioned in Section 2.2, cooling and heating zones behave differently in the designs of this study. The cooling zone (annealing) works by passive cooling, a process that cannot be accelerated. On the contrary, in the heating zones (denaturation and extension), the supplied heat can be increased, which can lead to more direct transitions from one temperature zone to the next and thus increase the residence time in each zone. This is the reason why in the third microreactor geometry, annealing and denaturation zones have almost equal residence times (Figure 6B), although the former covers a much larger area of the chip. The third microreactor geometry also gives the best results in terms of minimal time loss. Indeed, the functional time ratio (calculated from the total residence time in all steps, divided by the cycle time) of the third microreactor geometry is 74.7% (meaning that only 25.3% of the total time is lost during ineffective temperature transitions), which is higher, compared with the functional time ratios of 69.3% and 62.5% for the first and second microreactors.

### 3.2. Fabrication of the Microreactor and Comparison to Temperature Measurements

Figure 7 shows the design of the PCB layers for the third (best) microreactor geometry. In the center of the microreactor, the 20 mm diameter circular hole (yellow) can be seen. The microreactor was designed on four copper-layer PCBs. The microheaters were designed in a meandering shape and were located on the two inner copper layers of the PCB (Figure 7A, layer 2 (violet) and layer 3 (yellow)). As discussed in Section 2.2, the microheaters were formed only in the denaturation and extension temperature zones, where heating of the sample is required. At the beginning of each temperature zone, the microheaters are formed on two copper layers, so as to provide more heat to the specific areas, in order to achieve faster the desired temperature in the sample. On the contrary, in the middle of each temperature zone, the thermal elements are formed only on one copper layer, thus delivering less heat, so that eventually the temperature inside each zone is kept as stable as possible and close to the desired value. At the fourth copper layer of the PCB, above the microheaters and below the area where the circular microchannel is to be formed, three solid copper layers exist, one for each temperature zone (Figure 7B, layer 4 (red)). These three solid copper layers are placed in order to improve the temperature uniformity in each zone. In between the copper layers, the insulation holes of elliptical shape can be seen (Figure 7, yellow color).

Figure 8A shows the PCB chip that was fabricated, based on the design in Figure 7. The integrated microheaters are internal, while the solid copper films are covered by the solder mask used in the PCB industry for protection (more information on the stack can be found in Appendix A). The three separate temperature zones can be seen, separated by the elliptical through holes that provide thermal insulation. The power supply points of the chip are also visible, as well as the points of vertical interconnection access (VIA), where the different copper layers are connected to each other, in order to complete the electrical circuit.

Following the fabrication of the chip, a comparison was made between the temperatures achieved on the fabricated chip and the predicted temperatures of the computational study. Figure 8B shows the temperatures measured at the surface of the fabricated PCB chip with the use of the thermal camera. These temperatures were monitored over a period of 30 min, during which they exhibited exceptionally small fluctuations, indicating that the chip can achieve a constant temperature distribution over time, a requirement for successful and reproducible PCRs.

Because the fabricated chip has no microchannel constructed on its surface, a new computational study was conducted for the same microreactor (third microreactor geometry) but this time without a microchannel or sample circulation (more information can be found in Appendix A). This new computational study uses the same heat generation rates ($P_{Den}=0.771\;W$ and $P_{Ext}=0.646\;W$) (see Appendix A) that were used to obtain the optimal performance for the third microreactor. This means that the microheaters produce the same amount of heat as they did in the previous computational study with the sample circulation. Figure 8C shows the temperatures across the surface of the chip that were calculated at a steady state in this new computational study. As expected, the new temperatures have a larger deviation between minimum and maximum values (compared to Figure 5C), as this time there is no sample circulation, and thus no convective heat flows through the different temperature zones of the microreactor. We should note that the temperatures, in both Figure 8B,C, are not the temperatures of the circulating sample (these are shown in Figure 5C and Figure 6A). Instead, the presented temperatures are of the surface of the chip, when heated without sample circulation. When sample circulation is added to the chip, the moving sample stays for a very short time above each temperature zone and thus does not have the time to reach the temperatures in Figure 8B,C.

A comparison between the temperature profiles obtained from the computational study and from the fabricated chip is presented in Figure 8D. The data were collected along a circle of 15 mm in radius which is located exactly where the microchannel is going to be fabricated and is shown in Figure 8B,C with dashed lines. Two monitoring points (shown in Figure 8C) were used for the comparison between the results of the computational study and the experimental results. The two monitoring points (MPs), one for each independent microheater, were selected to be in the denaturation zone (MP1) and in the extension zone (MP2), as the temperature in each of these zones can be adjusted by changing the thermal power of the respective microheater. More specifically, the electrical power provided to each microheater was set so that the temperatures at the two monitoring points would have exactly the same values as the ones calculated from the computational study. This can be seen in Figure 8D, where the temperatures obtained from the computational study and experimental results coincide at the two monitoring points ($T_{MP1}=106.1\;{}^{\circ}C$ and $T_{MP2}=80.9\;{}^{\circ}C$). The aim of this comparison was to see (a) whether or not the two temperature profiles would be close to each other after having adjusted the temperatures of the two monitoring points to match and (b) if the thermal power that was used in the PCB chip would be relevant to the heat generation rate that was used in the computational study.

To obtain the temperatures presented in Figure 8B, a voltage of $V_{Den}=5.1\;V$ and $V_{Ext}=4.9\;V$ was applied to the microheaters of denaturation and extension, respectively. A time period of approximately one minute was needed for the temperature zones to reach the constant values presented in Figure 8B. These values combined with the resistance values of $R_{Den}=31.08\;\Omega$ and $R_{Ext}=35.07\;\Omega$ for denaturation and extension microheaters while heated give us a power consumption of $P_{Den}=0.837\;W$ and $P_{Ext}=0.684\;W$, resulting in a total power consumption of $P_{Tot}=1.521\;W$. The latter indicates a low power footprint for heating the microreactor, reinforcing the portability potential of the device. Of course, power consumption for sample circulation and amplified DNA detection should be included in the total power balance. The experimentally measured power consumption for heating is only ~7.3% (0.104 W) larger than the heat rate generation that was calculated in the computational study (1.417 W). This small difference may be due to contact resistance not taken into account in the computational study. It can be noticed in Figure 8D that the two temperature profiles are very close to each other, with the maximum difference not exceeding 4 °C and the average temperature difference being $\Delta T_{av}=0.256\;{}^{\circ}C$.

The excellent agreement of the model results with the measured temperature profiles and the consumed power emphasizes the reliability of the model as well as the validity of the values used for the thermophysical properties. The results also demonstrate that a CL microreactor with a circular microchannel can potentially operate in very fast PCR cycles with low energy cost, which is a big step toward ultra-fast, portable, and lab-independent μPCRs. Some challenges still remain to be addressed for the operation and full validation of this type of microreactor, with the most important one being the means for sample circulation in a closed-loop microchannel. Although some solutions have been already proposed to address this issue [63], this matter needs further investigation. Both ferrofluid-driven circulation (as in previous works [15,50,51]) and an on-chip peristaltic flow mechanism will be explored.

## 4. Conclusions

In this work, the performance of three different closed-loop μPCRs, designed with a circular microchannel to be fabricated on the PCB substrate was evaluated by means of a computational study. In order to evaluate the performance of each microreactor, residence times for all PCR steps were calculated, targeting the fastest possible PCR cycle time and the performance of a PCR protocol close to 1:1:2.

Subsequently, a PCB chip was fabricated based on the design of the microreactor that performed best in the computational study. The fabricated PCB chip, which at the time of this writing, is the first CL μPCR microreactor fabricated on a PCB, was tested by measuring the temperature profile on its surface with the use of a thermal camera, without sample circulation. These results were then compared with the ones of the computational study, in order to evaluate the reliability of the latter, and excellent agreement was demonstrated.

CL μPCRs with a circular microchannel can offer many important improvements over most existing CF μPCRs, most noticeable being the smaller footprint, flexibility in the total number of PCR cycles to be performed, better temperature reproducibility between the PCR cycles, and a lower pressure drop in the microchannel during operation.

At the same time, the PCB constitutes a great substrate compared with other materials, as it gives the possibility of having integrated microheaters inside the chip (permitting thus low-power consumption), as well as other electronic components (such as sensors). The techniques required to manufacture the PCB chips are already widely used by the consumer electronics industry, which allows for low-cost, reliable, reproducible, mass-amenable fabrication.

Combining the advantages of both CL μPCRs and PCBs, it becomes clear that the proposed microreactor shows great potential and constitutes an excellent component of a lab-on-chip system used for POC testing. The results of the computational study show that, for a sample of ~10 μL and a DNA target of 128 bp, the proposed design (microreactor 3) could potentially allow for ultra-fast amplification of DNA, although an actual PCR run has not been implemented yet. More specifically, the study suggests that for the aforementioned design, a PCR cycle time of 5.7 s can be achieved, which would lead to a total PCR time of less than 3 min, for 30 PCR cycles. The total power consumption of the microreactor is measured to be $P_{Tot}=1.521\;W$, and the thermal energy required for 30 cycles is 260.1 J. These values suggest that the heating of the microreactor can be achieved with the use of a portable battery, an advantage for POC systems. The speed and the low power requirements, when combined with the low-cost fabrication and the flexibility of CL PCR microreactors, provide a very promising solution for POC systems. The current study holds as the first and proof-of-concept step of ongoing work to realize ultra-fast PCR in the microreactor, combined in the future with an optical system for performing real-time quantitative PCR.

## Figures and Tables

**Figure 1 micromachines-14-00172-f001:**
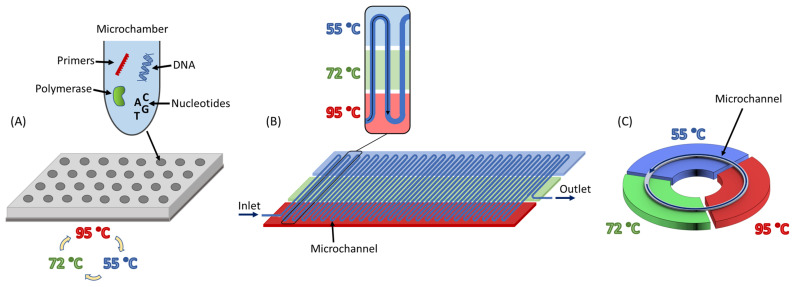
(**A**) Time-domain μPCR. The PCR sample is placed inside microchambers where the temperature is cycled. (**B**) Space-domain, continuous-flow (CF) μPCR. The sample is pumped through a microchannel, passing through zones of constant temperatures. (**C**) Space-domain closed-loop (CL) μPCR. The sample flows through a circular microchannel, passing consecutively and repetitively through the different temperature zones.

**Figure 2 micromachines-14-00172-f002:**
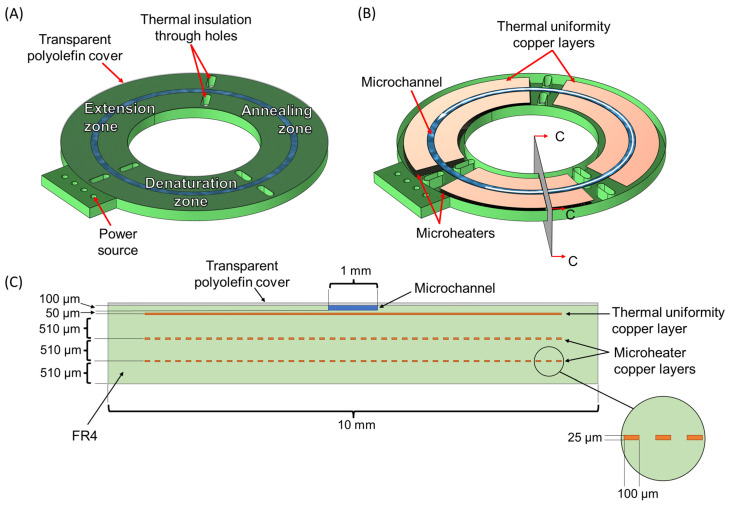
CL microreactor geometry. (**A**) External view of the microreactor with the circular microchannel covered by a thin polyolefin film. (**B**) Internal view of the microreactor. The sample is flowing through the microchannel following a counterclockwise direction (white arrow) along the three temperature zones of the PCR. (**C**) Cross-section of the microreactor with the C-C plane. The microheaters and thermal uniformity copper layers are integrated.

**Figure 3 micromachines-14-00172-f003:**
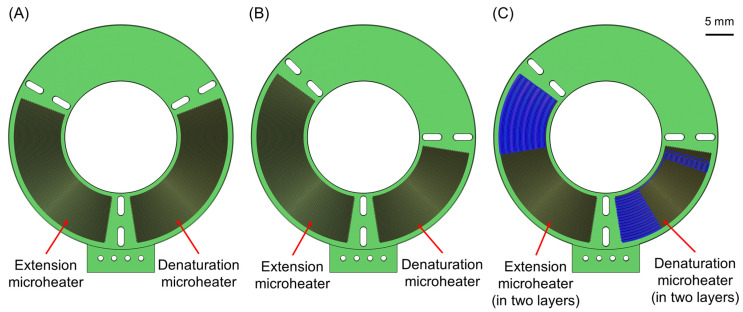
The three different microreactor geometries. (**A**) First microreactor geometry, with the temperature zone areas evenly distributed on the surface of the chip. (**B**) Second microreactor geometry, with the denaturation zone reduced in favor of the other two temperature zones. (**C**) Third microreactor geometry, with the ratio of the temperature zones as in the second geometry but with the microheaters designed in two different copper layers inside the PCB.

**Figure 4 micromachines-14-00172-f004:**
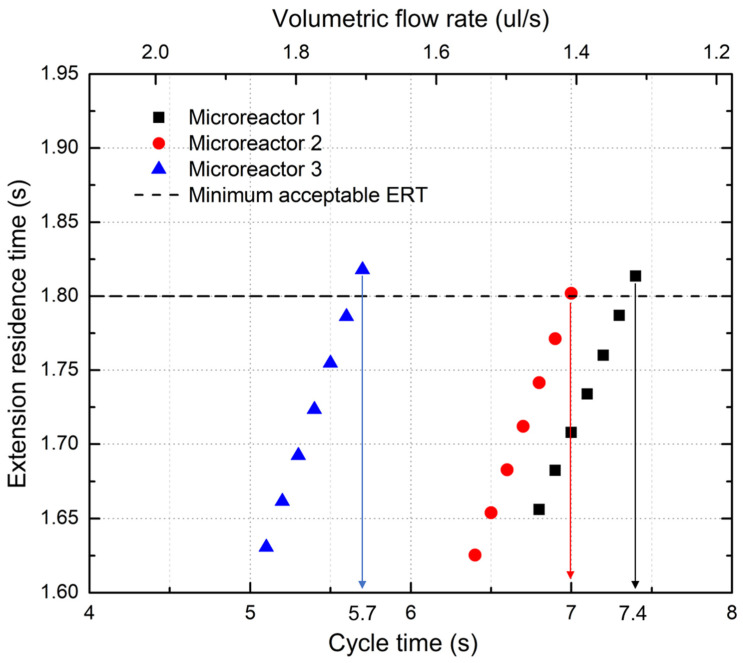
Extension residence time vs. cycle time (or volumetric flow rate) for all three microreactors.

**Figure 5 micromachines-14-00172-f005:**
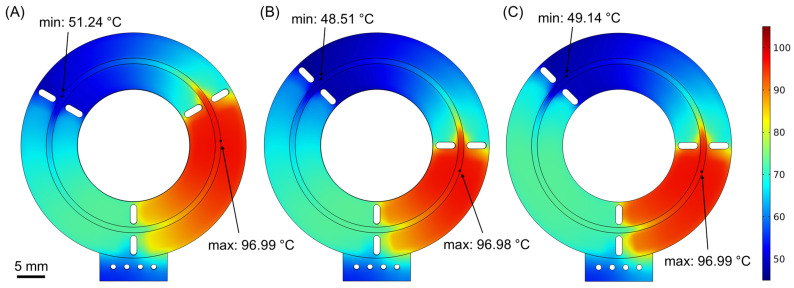
(**A**–**C**) Temperature distributions (at the middle height of the microchannel) for the three microreactor geometries that are shown in Figure 3, as calculated in the computational study. Each microreactor geometry operates at the fastest possible flow rate ($\dot{V}$_1_ = 1.26 μL/s, $\dot{V}$_2_ = 1.31 μL/s, $\dot{V}$_3_ = 1.65 μL/s), with the cycle times being 7.5 s, 7.2 s, and 5.7 s, respectively. Maximum and minimum values refer to the temperatures of the sample inside the microchannel.

**Figure 6 micromachines-14-00172-f006:**
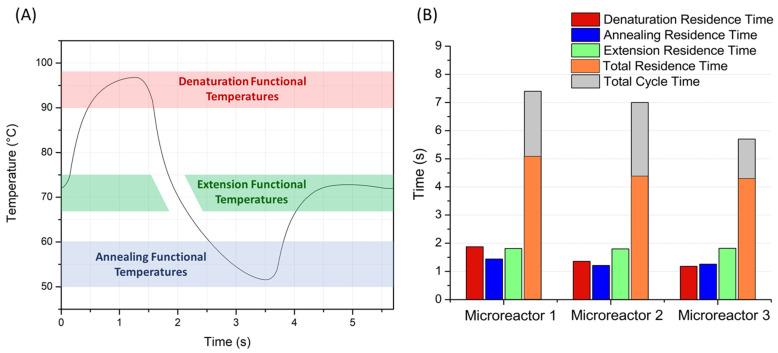
(**A**) Temperature profile taken at the middle of the microchannel, as a function of time, for the third microreactor geometry (representative case). The colored areas show the functional temperature ranges, where the three PCR steps can take place efficiently. (**B**) Residence times and total cycle times for the three microreactor geometries.

**Figure 7 micromachines-14-00172-f007:**
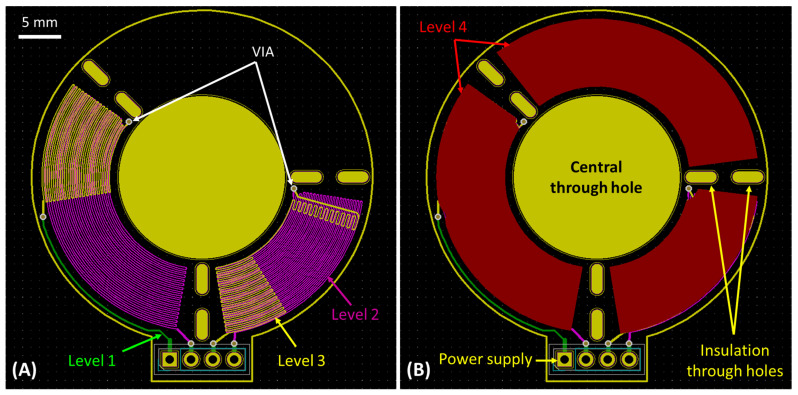
The geometry that achieved the best results in the computational study (3rd microreactor) was designed for fabrication in PCB, with the use of KiCad software program (5.1.0). On the left (**A**), the integrated microheaters can be seen at copper layers 2 (violet) and 3 (yellow) of the PCB. In the first and lower copper layer (green), the microheaters are connected to the temperature controller. The layers are interconnected at specific points with vertical interconnect access (VIA) elements. On the right (**B**), three solid copper layers are shown in red. These copper layers are located above each temperature zone (layer 4) to provide improved temperature uniformity.

**Figure 8 micromachines-14-00172-f008:**
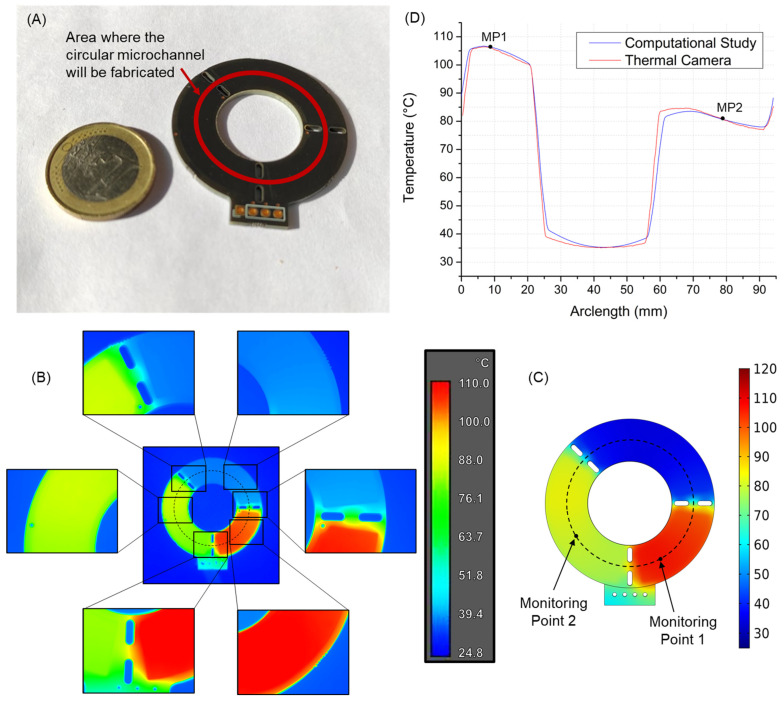
(**A**) PCB chip fabricated on the basis of the design in Figure 7. (**B**) Temperature distribution across the surface of the fabricated PCB chip, as measured with the use of a FLIR A300 thermal camera (image in the middle) and an external lens that provided magnification (side images). (**C**) Temperature distribution across the surface of the fabricated chip (third microreactor geometry) without microchannel or sample flow, as calculated in the computational study. The two monitoring points can be seen, in the denaturation and extension zones. (**D**) Comparison of the temperature profiles calculated in the computational study and measured on the fabricated chip surface, along a circle with radius of 15 mm (seen with dashed lines in (**B**,**C**)). The two monitoring points can be seen in the denaturation and extension zones.

**Table 1 micromachines-14-00172-t001:** Thermophysical properties of the materials (see Figure 2). Not applicable material properties, such as the emissivity of water which is enclosed within the geometry, are marked as n/a. FR4 thermal conductivity is given in the form of a matrix because its value changes depending on the direction of the heat transfer. The values of thermal conductivity, heat capacity at constant pressure, and density of copper and water depend on the temperature and are taken from the library of the computing software.

Property	Units	Material				
		Polyolefin	FR4 (PCB)	Copper	H_2_O	Solder Mask Polymer
Thermal conductivity	$\frac{W}{m\cdot K}$	0.12	$\left[\begin{array}{ccc} 0.81 & 0 & 0\\ 0 & 0.81 & 0\\ 0 & 0 & 0.29 \end{array}\right]$	$k_{copper}\left(T\right)$	$k_{H_{2}O}\left(T\right)$	0.25
Heat capacity at constant pressure	$\frac{J}{kg\cdot K}$	$1.09\cdot 10^{-3}$	1369	$C_{copper}\left(T\right)$	$C_{H_{2}O}\left(T\right)$	1100
Density	$\frac{kg}{m^{3}}$	1400	1900	$\rho_{copper}\left(T\right)$	$\rho_{H_{2}O}\left(T\right)$	1.9
Surface emissivity	–	0.95	0.9	0.2	n/a	0.99

## Data Availability

Not applicable.

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
