# Peer review of "Closed-Loop Microreactor on PCB for Ultra-Fast DNA Amplification: Design and Thermal Validation"

_micromachines, 2023, doi:10.3390/mi14010172_

Round 1

Reviewer 1 Report

The performance of three distinct CL PCRs created for manufacturing on a printed circuit board (PCB) was assessed in this publication using computational research in terms of the residence duration in each heat zone. The residence periods were retrieved from the microreactor's temperature distribution, which was calculated using a 3D heat transfer model. On the basis of the design that performed the best in the computational analysis, a PCB chip was subsequently created. The constructed PCB chip, which was claimed as the first CL PCR microreactor made on PCB, was evaluated by monitoring the temperature profile on its surface using a thermal camera without sample circulation. In this study, it was demonstrated that the PCR cycle time of 5.7 s, which produced a total PCR duration of less than 3 min, for 30 PCR cycles, could be achieved. The demonstrated approach could be useful for lab-on-chip sample analysis, and thus is valuable and relevant to the journal. The manuscript is well-written and the results support the claims of the authors. I have the following comments

* What are the benefits of the circular channel over the existing more commonly used designs?

*What is the limitations related to the microchannel hights? (how will the fluidic channel height change the cycle durations?)

*some estimates on the cost of the system could be useful.

*It could be better to include some important figures of merit in the abstract. 

Reviewer 2 Report

In this manuscript, the authors compared the performances of three different closed-loop μPCRs through a numerical study. Then, based on the design that performed best in the computational study, a PCB chip was fabricated to evaluate the reliability and to verify the accuracy of the developed model. The obtained results showed that better temperature reproducibility between the PCR cycles, faster cycle speed and lower power consumption have been achieved. It provides a credible way to realized ultra-fast PCR in the microreactor. So, I recommend that this manuscript can be published after some minor revision.

1. The formatting of references, including Issue number and volume number, need to be doublechecked. e.g., citation 10, citation27...;

2. Scale bars should be added to the diagrams of the models;

3. More detailed information, such as model setting and operating methods are suggested to supplement in Section 2.

Reviewer 3 Report

See Word file

Round 2

Reviewer 3 Report

The article has been well adapted and can be published if the following minor feedback is adapted:

1. R489: relatively and closer is double

2. Although the title of the article already suggests it, but I would state more clearly in the abstract that no actual PCR has been carried out

3. It is still not fully clear how the sample is moved through the reaction zones. I do understand the author's opnion to not focus to much on this in this study. But since the method of sample movement is of importance for the chip design, I recommend to say a few more words on this. 

4. Please give some more details about the costs. 

5. The explenation on how to achieve the 1:1:2 protocol (as closely as possibble) is clear. Please also add this to the article. 

6. R150: formed on or in the PMMA substrate? --> I agree it can be both. But I got no answer on what was used in this study.
